# Targeted Mutagenesis of the Female-Suppressor *SyGI* Gene in Tetraploid Kiwifruit by CRISPR/CAS9

**DOI:** 10.3390/plants10010062

**Published:** 2020-12-30

**Authors:** Gloria De Mori, Giusi Zaina, Barbara Franco-Orozco, Raffaele Testolin, Emanuele De Paoli, Guido Cipriani

**Affiliations:** 1Department of Agricultural, Food, Environmental and Animal Sciences, University of Udine, Via delle Scienze 206, 33100 Udine, Italy; giusi.zaina@uniud.it (G.Z.); barbara.franco@tdea.edu.co (B.F.-O.); raffaele.testolin@uniud.it (R.T.); emanuele.depaoli@uniud.it (E.D.P.); guido.cipriani@uniud.it (G.C.); 2Facultad de Ingeniería, Tecnológico de Antioquia–Institución Universitaria TdeA, Calle 78b No. 72A-220, Medellín-Antioquia 050001, Colombia

**Keywords:** *Actinidia* spp., sex-determinant, hermaphroditism, plant transformation, genome editing, new breeding technologies (NBTs)

## Abstract

Kiwifruit belong to the genus *Actinidia* with 54 species apparently all functionally dioecious. The sex-determinants of the type XX/XY, with male heterogametic, operate independently of the ploidy level. Recently, the SyGI protein has been described as the suppressor of female development. In the present study, we exploited the CRISPR/Cas9 technology by targeting two different sites in the *SyGI* gene in order to induce a stable gene knock-out in two tetraploid male accessions of *Actinidia chinensis* var. *chinensis.* The two genotypes showed a regenerative efficiency of 58% and 73%, respectively. Despite not yet being able to verify the phenotypic effects on the flower structure, due to the long time required by tissue-cultured kiwifruit plants to flower, we obtained two regenerated lines showing near fixation of a unique modification in their genome, resulting in both cases in the onset of a premature stop codon, which induces the putative gene knock-out. Evaluation of gRNA1 *locus* for both regenerated plantlets resulted in co-amplification of a minor variant differing from the target region for a single nucleotide. A genomic duplication of the region in proximity of the Y genomic region could be postulated.

## 1. Introduction

The kiwifruit belongs to the genus *Actinidia*, which comprises 54 species characterized by climbing perennial plants mainly found in Southern China [1,2]. A characteristic trait of this genus is the dioecism, meaning that male and female flowers are borne by different plants. Male individuals bear staminate flowers with numerous stamens producing viable pollen and a rudimentary ovary lacking ovules, while female individuals bear pistillate flowers endowed with a well-developed ovary and hundreds of ovules, and stamens with functionally sterile anthers [3,4,5]. Kawagoe and Suzuki [6] proposed that the stamens of pistillate flowers aid reproduction by attracting pollinating insects. All *Actinidia* species are apparently functionally dioecious, although dioecism has been confirmed in only a few species [6]. Dioecism is not absolute, and inconstant males, also referred to as fruiting males, were identified among the pool of male pollinizers in commercial kiwifruit orchards and germplasm repositories [3,7]. These individuals have bisexual flowers characterized by a small ovary with fewer carpels than a typical female vine, fewer ovules per carpel, and shorter, thinner styles with small stigmata [3], and they bear small fruits (20–40 g) with a few dozens of seeds. However, a labile and inconstant sex expression with staminate, bisexual, and mixed inflorescences was observed in a mosaic pattern [7].

Dioecism brings inevitable disadvantages in kiwifruit breeding. Since male plants do not bear fruit, paternal parents for crosses are selected with unknown fruit quality [8,9]. Moreover, in a typical cross-population, dioecism results in a final female to male ratio of 1:1, with male plants representing a waste of land and resources provided that breeders are usually interested in selecting fruit-bearing individuals. Those hindrances are exacerbated by the long generation cycle. Development and introduction of improved cultivars by plant breeders may require many breeding cycles and dozens of years [10]. The management cost of relatively large cross populations over a number of years is a limiting factor in any woody species breeding program, particularly in fruit species like kiwifruit, where expensive support structures are necessary.

McNeilage and Steinhagen (1998) [11] identified a natural hermaphrodite, classified as a rare variant within the progeny derived from a cross having an inconstant male as parent. Despite this, hermaphroditism in kiwifruit is an extremely erratic character and breeding cannot be based on such occurrences. On the other hand, hermaphroditism is considered the Holy Grail of the modern kiwifruit industry. Approximately 12% of the canopy in a kiwifruit orchard is occupied by male pollinizers with a comparable loss of productivity [12]. Land use and pollinizer management is not the sole cost for farmers. Good pollination is essential to produce fruits of good size, as fruit development is dependent on seed content [12]. Bee-hives are often placed into a kiwifruit orchard in order to increase fruit set and fruit size [13]. The alternative, artificial pollination with pollen produced in dedicated male orchards, has been recently criticized because of concerns about the diffusion of bacterial canker disease caused by *Pseudomonas syringeae pv. actinidieae* [14].

In *Actinidia*, sex-determining genes are localized in a pair of karyotypically indistinguishable chromosomes, which function like an XX/XY system with homogametic females (XX) and heterogametic males (XY). Individuals carrying at least one copy of the Y chromosome are males, regardless of the ploidy level, suggesting that the sex-determining *locus* includes an active male-determining factor [15]. The suppression of the recombination around sex-determining genes is the key feature of sex chromosomes, allowing the two chromosomes of a pair to evolve separately and the multiple genes involved to be inherited as a single genetic determinant so that only male and female progeny are produced [16]. Recently, Akagi and co-workers described the two Y-encoded sex determinants in kiwifruit: the Shy Girl (*SyGI*) protein, which is a Y-encoded cytokinin response regulator that acts as the suppressor of female development [17], and Friendly boy (*FrBy*), a fasciclin-like protein which would maintain male fertility [18]. They demonstrated that *SyGI* gene is expressed in developing flowers, specifically at the surface of the rudimentary carpels of male flowers, while *FrBy* exhibits strong expression in tapetal cells. Moreover, *FrBy* acts for the maintenance of male (M) functions, independently of *SyGI*.

In the present study, we exploited the clustered regularly interspaced short palindromic repeat (CRISPR)/CRISPR-associated protein 9 (Cas9) system to knock-out the *SyGI* gene and induce stable hermaphroditism in a male genotype of *Actinidia chinensis* var. *chinensis*. Creation of mutants is a tool in investigating gene function and crop improvement [19], and targeted genome editing approaches mediated by site-specific nucleases like Cas9 are precise, convenient, and time-saving compared to traditional breeding approaches. Since generating new kiwifruit selections using conventional breeding programs is generally tedious and time-consuming, the recently developed CRISPR/Cas9 system provides a favorable opportunity for perennial crop improvement, based on its highly efficient first generation genome editing scheme [20].

Compared to other genome editing approaches [21], the CRISPR/Cas9 system is particularly useful for multiplexed gene editing, because its target specificity is based on short and single guide RNA molecules (sgRNA). Some studies have already been conducted in the diploid *Actinidia chinensis* female cultivars “Hongyang” [22] and “Hort16A” [18,23]. In this study, we used a CRISPR/Cas9 multiplexing vector [24] and two guide RNAs (sgRNA1 and sgRNA2) to target the gene *SyGI*. The assembled vector was cloned into *Agrobacterium tumefaciens* LBA4404 and EHA105 strains and used to infect leaf discs and petioles of two *Actinidia* male accessions in order to obtain stable transformants. We used two different *A. chinensis* tetraploid male genotypes frequently used as pollen donors in our breeding program. The same disomic sex segregation ratio seems to operate at different ploidy levels in *Actinidia*, and a single Y chromosome is expected to be involved in all cases in sex determination [15].

## 2. Results

### 2.1. SyGI Target Identification

Cas-Designer tool (see Section 4.2) was run using the two exons of *SyGI* gene sequence as input in order to achieve all the possible targets with the relative information, including potential off-targets within 2-nt mismatches and optional 3-nt bulge. Cas-Designer provided 37 putative targets on the first exon and 12 putative targets on the second one. Among those, we selected the two sgRNAs that showed the best values for GC content and out-of-frame score, and a score equal to zero in the mismatches evaluation (Off-target sites) throughout the *Actinidia chinensis* genome. Moreover, the two best-performing sgRNAs were selected so that they were located in the two different exons, in order to maximize the probability of a successful editing through the creation of a premature stop codon and/or a large deletion. The sgRNA1 (ACAAAAATGGCCCGGCAACA) mapped on the first exon (35 nucleotides apart from the exon start position), while sgRNA2 (TGTGAAGCCGTTGACTGCTG) was in the second exon (5′ position at nucleotide 53 of the exon) in the reverse strand, spanning a distance of 1651 nucleotides from one to the other.

Cas-OFFinder (see paragraph 4.2) gave no off-target site in our genome for the two selected guides. However, it is possible for Cas9 to recognize slightly different sequences forming a DNA or RNA bulge. One such sequence for the sgRNA1 is present in the *Actinidia chinensis* genome in the minus strand of LG19 in position 5,058,376, within the sequence of *Achn384741*, which is the autosomic counterpart of our gene of interest [17]. That sequence displays two mismatches in the seed part of the guide, making Cas9 binding highly unlikely. However, the possibility of a Cas9 mismatch was taken into consideration. As for the sgRNA1, the choice was also based on the presence of a restriction site for the *Bsl*I enzyme within the predicted cleavage site, as shown in Appendix A, which can be exploited to set up a screening assay of the transformed plants to verify the success of the editing.

### 2.2. CRISPR/Cas9 T-DNA Vector Assembly

The final p*DIRECT22C*:gRNA vector (Figure 1) was sequenced through NGS Illumina technology and produced a total of 947,032 reads corresponding to a coverage of 60X, which was afterwards assembled in a unique scaffold. The original p*DIRECT22C* vector was 16,058-bp long, while the vector we assembled consisted of 15,552 bp after the insertion of the guides and the loss of the *ccdB* gene, as expected by the cloning procedure. The alignment of the assembled vector sequence and the original one showed a similarity of 99.99% between the two sequences. Only two SNPs were detected, both located in a non-coding region of the vector backbone.

### 2.3. Actinidia Transformation

Two different types of plant explants, leaf discs and petiole segments, of the two male kiwifruit genotypes A0134.41 and Ac174.46 were used for the stable transformation through *A. tumefaciens* LBA4404 strain harboring the p*DIRECT22C*:gRNAs vector.

We infected 40 leaf discs and 35 petioles of A0134.41 and 50 leaf discs and 33 petioles of Ac174.46. The genotype A0134.41 reacted positively to the *A. tumefaciens* LBA4404 strain inoculation, providing viable plant tissues with regeneration cores. On the other hand, Ac174.46 did not, with the plant material showing extensive necrosis, leading to the final death of the whole explant. Ac174.46 had been previously tested for its regenerative capability (data not shown) and showed a good regeneration rate when grown on a non-selective medium. Consequently, given the plant genotype specificity of *Agrobacterium* infection efficiency, we decided to use a different *A. tumefaciens* strain (EHA105) to infect Ac174.46.

Kanamycin-resistant calli appeared along the cut edge of the explants in 4–8 weeks (Figure 2A). The number of calli varied according to the type of starting material and the different genotypes. In both cases, petioles gave consistent results, while leaf discs did not. We obtained only two calli from the leaf discs of the A0134.41, while, on average, 60% of petioles from A0134.41 and 20% from Ac174.46 produced kanamycin-resistant calli.

The first kanamycin-resistant adventitious buds were harvested from the calli approximately 12–16 weeks after explants had been co-cultivated with *Agrobacterium* (Figure 2B). Other kanamycin-resistant adventitious buds regenerated after 2–3 cycles of transfer onto fresh regeneration medium. We collected 44 putative transgenic shoots for A0134.41 and 61 for Ac174.46, which were in-vitro propagated for further analyses (Figure 2C). Overall, after transformation the two genotypes showed a regenerative efficiency estimated at 58% in the case of AO134.41 and 73% in the case of Ac174.46, as detailed in Table 1.

### 2.4. Evaluation of Mutation Induced by Editing

The genomic analyses to assay editing results were conducted on 11 putative transgenic shoots obtained from the male genotype A0134.41 and four putative transgenic shoots obtained from the male genotype Ac174.46.

First, T-DNA integration into the recipient plant genomes was tested by amplifying the insert carrying the sgRNA transcriptional cassette within the p*DIRECT22C*:gRNAs vector. Genome integration was confirmed for seven out of 11 putative transgenic shoots of A0134.41 and two out of four putative transgenic shoots tested for Ac174.46 (Figure 3).

Then, the presence of editing within *SyGI* sequence targets defined by sgRNA1 and 2 was screened using two independent approaches. The first, based on the T7EI assay using T7 endonuclease (Rif.), was expected to yield cleavage products in presence of a mixture of edited and unedited DNA products but did not provide evidence for such condition in any sample (data not shown). As this result could be ascribed to either pervasive DNA modification throughout the sample DNA mixture, editing failure, or sensitivity issues, definitive evidence for editing was provided by the second approach, which was based on site-specific DNA digestion. Indeed, we exploited the presence of a restriction-enzyme site in sgRNA1 target sequence to set up a screening assay of the transformed plants. The restriction assay is based on the presence of the *Bsl*I restriction site in correspondence to the predicted cleavage site of the Cas9 enzyme in sgRNA1 target sequence.

The amplicon used in the restriction assay is 427-bp in length and displays two *Bsl*I restriction sites within the *wild-type* sequence at position 222 nt (the predicted cleavage site defined by sgRNA1) and 320 nt, producing three fragments of 221, 98, and 108 bp after enzyme digestion. In the case of an edited plant, the cutting site at position 222 is supposed to be modified by the editing, providing a different digestion pattern (two fragments of 320 and 108 bp). The *Bsl*I assay has been used to test the 15 putative transgenic plants (Figure 4), and a summary of the assay results has been reported in Table 2. Thirteen of these plants exhibited a restriction pattern compatible with the wild type, showing the fragment of 221 bp in length. In contrast, two different transgenic shoots, named A0134.41_L3 and Ac174.46_L1 (in bold in Figure 4), showed an electrophoresis pattern with only two fragments (320 and 108 bp in length), consistent with pervasive DNA modification in all the cells of these samples or in most of them, considering the limits of sensitivity of this technique. Based on this preliminary information, we set out to carry out a definitive evaluation and characterization of the editing products in a subset of selected plants by amplicon sequencing of the genomic targets with Illumina technology.

Two genomic fragments, 427 and 432 bp in length, centered around the sgRNA1 and sgRNA2 targets, respectively, were PCR amplified and sequenced at high processivity from leaf tissue of eight plants: the two plants showing pervasive editing in the DNA restriction analysis, four plants with evidence for partial editing, and two plants that apparently failed to be edited (Table 2). In all these samples, sequencing reads from the sgRNA1 target region showed proof of co-amplification of a duplicated genomic *locus*, differentiated from the bona fide target region by eight noncontiguous single nucleotide changes and a 10-nt deletion. Two of these SNP positions were located within the expected sgRNA target site, homologous to the sgRNA1 intended target, possibly preventing the interaction between DNA and the sgRNA. Indeed, reads assigned to this template did not exhibit any editing in the expected position and were disregarded from following analyses (see also Discussion).

The reads unambiguously assigned to the intended target region of the *SyGI* gene amounted to approximately 40,000–50,000 per sample and were further sorted in two variants based on a A to T substitution located in the upstream primer region at 22 bp from the amplicon 5′ end in the scaffold orientation for the Ac174.46 genotype or a A to C substitution 10 bases downstream of the primer region (31 bp) in the case of the second edited genotype, A0134.41. The relative positions to the *SyGI* gene reference scaffold are, respectively, 45,746 and 45,755 bp.

Except for sample Ac174.46_L1, where reads were equally sorted between the two variants, in all the other sequenced samples one variant was present in about 10% of the reads. Thus, in the event that the two variants were allelic (see Discussion for observations indicating that this hypothesis is unlikely), PCR amplification and sequencing did not provide the expected balanced representation. Nevertheless, editing statistics were based on the total amount of reads assigned to the *SyGI* gene, irrespective of the sequence variant, as both variants presented evidence for editing when this occurred.

In-silico sequence processing by the CRISPResso software highlighted that the two regenerated lines named A0134.41_L3 and Ac174.46_L1, which appeared pervasively edited according to the DNA restriction analysis, show nucleotide modification at the target site in 99.85% and 99.76% of total assigned reads, respectively (Figure 5). In the case of A0134.41_L3, the type of editing observed is an insertion of an A in the point of the predicted cleavage site, while in the case of Ac174.46_L1, we observed a deletion of five bases and an insertion of a G in the same position. Both modifications are predicted to cause a shift in the reading frame during protein synthesis and generate premature stop codons downstream of the cleavage site, as shown in Figure 6. Therefore, in both regenerated lines, the editing proved to be effective in producing the changes required for gene knock-out. In all the other plants that were analyzed, the fraction of edited reads varied between 0% and 0.31% (see Table 2 for details), indicating that the DNA restriction analysis was also correct in identifying true unedited plants, whereas the chimeric or mosaic condition postulated for some plants based on that approach resulted to be associated with negligible amounts of edited cells if any.

The two lines A0134.41_L3 and Ac174.46_L1 have been maintained in climatic chamber on maintenance medium (Appendix A) and propagated for further analysis (Figure 7B,C).

## 3. Discussion

As previously stated, all *Actinidia* taxa appear to be functionally dioecious, although this has been unequivocally established in only a few taxa such as *A. chinensis* and *A. polygama* [6,13]. A single Y chromosome appears to be sufficient for maleness at any ploidy level [4,15,25]. Therefore, in diploids, females would be XX and males XY, in tetraploids the females XXXX and the males XXXY, in hexaploids the females XXXXXX and the males XXXXXY, etc. In kiwifruit, dioecism and intraspecific polyploidy are great challenges in breeding and cultivar improvement. Dioecism represents a huge problem for the presence of a 12% of unproductive pollinizers, but also for the complications in pollination due to asynchrony of blossoming between pollinizers and producing plants [26]. *Actinidia* flowers are not attractive to insects, and the pollen is neither sticky enough for insect pollination nor powdery enough for wind pollination.

Dioecism is also a problem from the breeding point of view. Indeed, in dioecious plants, the pollen is usually selected in the absence of knowledge on genetic background controlling the fruit trait, as fruiting characteristics are not expressed in males [27]. Progeny tests are the only available tool to evaluate pollen characteristics, but these tests are expensive and time-consuming. The possibility to develop a stable hermaphrodite cultivar would offer great advantages, such as significantly increased yields, as pollinizer plants would no longer be required, and a partial or complete solution to most pollination-related problems.

Nevertheless, conventional breeding systems are strictly dependent on existing natural allelic variation and are often accompanied by loss of fitness and genetic diversity [28]. A pioneering attempt to induce stable hermaphroditism in *Actinidia* through biotechnological approaches that overcome such shortcomings has been reported by Akagi et al. [17,18] and consisted of artificially introducing the *FrBy* open reading frame into a “rapid-flowering” *A. chinensis* diploid female cultivar with the purpose of expressing exogenous male-promoting sex determinant factor (Mfactor) and maintaining male fertility via proper tapetum degradation. However, beside cisgenetic approaches, genome editing strategies have also been explored, in particular through the increasing exploitation of the CRISPR/Cas9 system, which offers a robust and versatile toolkit for functional genomic research with straightforward applications to plant molecular breeding (reviewed in [29]).

In our study, we focused on the knock-out of *SyGI* gene through the application of the CRISPR/Cas9 system on the two tetraploid male kiwifruit genotypes A0134.41 and Ac174.46 (*A. chinensis* var. *chinensis*), which showed at the progeny test to induce early ripening and high fruit weight, respectively, in the offspring [30]. We are confident that editing of other male genotypes could broaden functional genomic research in *Actinidia*.

Indeed, although *SyGI* activity has been described as a major player in preventing ovule formation, little is known about the target genes involved in the mechanisms underpinning gynoecium development and possible pleiotropic effects on other floral organs [31]. By silencing *SyGI* gene, it may be possible in the future to assess the physiological effects on the development of plants, thus gaining a better understanding of its biological function. We took the first step in this direction by silencing *SyGI* gene in two tetraploid *Actinidia* genotypes. Following methods described in Wang et al. [22], we developed the CRISPR/Cas9 strategy using the vector p*DIRECT22C* [24], which is particularly useful for multiplexed gene editing, and we selected two sgRNAs targeting the first and second exon of *SyGI* gene in order to maximize both the editing efficiency and the size of the region targeted for mutation. Nevertheless, sequence analysis of target loci in regenerated plants revealed that only sgRNA1 was capable of inducing site-specific mutations. No editing event was associated to sgRNA2 target site, and no large deletion spanning the two guide targets was observed. That happened despite the careful selection of the guides according to the best recommendations for *in*-*silico* guide design by means of the Cas-Designer tool. Indeed, sgRNA2 displayed the presence of some structural elements (i.e., GC content between 40% and 60%, purine residues in the four last nucleotides) correlated to editing efficiency of the guide [32], together with the highest score at Cas-Designer analysis. In light of the unpredicted malfunction of this guide and the successful results attained with the sgRNA1 guide alone, the choice of a multiplexing approach to test more than one guide in a single experiment proved very convenient, although the purpose of applying a double guide was meant to maximize the chances for a disruptive mutation through the alteration of a longer DNA region.

CRISPresso analysis of sgRNA1 target *locus* conducted on eight regenerated plant lines highlights that two of these regenerated lines, named A0134.41_L3 and Ac174.46_L1, show near fixation of a unique modification in their genomes. More precisely, we observed that in the case of A0134.41_L3, the type of editing observed is an insertion of an A in the point of the predicted cleavage site defined by sgRNA1, while in the case of Ac174.46_L1, we observed a deletion of five bases and an insertion of G. Both modifications observed in the two selected plant lines cause a shift in the reading frame during the *SyGI* gene expression, resulting in both cases in the onset of a premature stop codon, which induces the putative knock-out of the gene. Unfortunately, at this stage we cannot verify the phenotypic effect of these two kinds of modification on the flower structure. Indeed, to assess the phenotypic evaluation of the flowers we must wait for a long time, because *Actinidia* spp. are characterized by a juvenile unproductive period. In natural conditions the growth cycle is spread between two seasons, interrupted by winter dormancy [33,34].

Both restriction and sequence analysis of the edited site were consistent with the expected targeted mutation. However, sequence analysis of the editing products revealed potential caveats in the editing procedure due to a convoluted duplication history for the *SyGI* gene in the *Actinidia chinensis* genome. The primers we used to amplify the editing site could isolate two major sequence variants of the target region. One variant represented the vast majority of the amplicons, accounting for approximately 98% of total amplicons in our experimental dataset. The second variant diverged from the other by eight SNPs and a 10-nt deletion. We noticed that such a sequence mapped at a high score on the *Actinidia chinensis* genome in the minus strand of LG19 in position 5,058,376 in correspondence with the *Achn384741* gene sequence, which is the autosomic counterpart of *SyGI*. Whereas the observed polymorphisms allowed us to easily sort and disregard these sequences from the editing analysis, the possibility of *Achn384741* gene acting as an off target in the editing procedure could not have been ruled out in retrospect. This has fortunately turned out not to be the case, because of the presence of SNPs in the seed part of the sgRNA sequence, making Cas9 binding unlikely. Somewhat exacerbating the complexity of sequence analysis, PCR amplification also coamplified minor nucleotide variants of the target region, this time differing for a single nucleotide position from each other. The presence of a second allele in a genomic region of the *SyGI* gene, characterizing the male genotypes, can be excluded considering that the postulated sex-determining system operates like an XX/XY system in tetraploid genotypes [4,5,35]. Although the presence of XXYY male genotypes could be advocated [36], we can exclude for both genotypes the presence of a second Y allele because of the segregation pattern of populations obtained crossing those genotypes with different female plants. A genomic duplication of the region could explain what we observed. There were no evidences of tandem duplication in our sequences; therefore, we are led to believe that the postulated duplication is located somewhere in the Y region of the chromosome 25. We do not have any evidence if the second postulated gene could be functional or silenced. If functional, we do not see any recombination in cross populations, reinforcing the hypothesis that the duplicated gene should be located in the proximity of the Y genomic region. A similar scenario has been described in poplar, where partial duplications of the *Arabidopsis Response regulator 17* (*ARR17*) orthologue in the male-specific region of the Y-chromosome were observed [37]. The *ARR17* gene, being an A-type cytokinin response regulator, shows functional similarity with the *SyGI* gene, but further analyses should be carried out to investigate whether a similar scenario could be present in the kiwifruit genome. In the event that the extra copy is complete and retains coding potential, the regenerated plants will probably be null mutants for both copies but the difference in sequence between the two copies would offer the future possibility of a selective knock-out.

In conclusion, the editing of the *SyGI* gene on the two tetraploid male kiwifruit genotypes A0134.41 and Ac174.46 allowed the selection of two different tetraploid plant lines, both showing the putative knock-out of the gene. We have tested only a few plantlets so far, but more transformed plants are still being cultivated in the elongation medium and we are fairly confident that the screening of those transformants will reveal more putative knock-out lines. However, as mentioned above, it will take some time to assess the phenotypic effects of the gene knock-out on flower development. To avoid that constraint, one possibility would be to simultaneously edit kiwifruit plants for the trait of interest and the CEN-like loci involved in growth habit and terminal flowering, exploiting the findings of Varkonyi-Gasic et al. (2019) [23]. This approach, adapted to a tetraploid genomic context, could allow acceleration of the phenotypic evaluation of traits associated to sex development.

## 4. Materials and Methods

### 4.1. Plant Material

Plant material from two tetraploid male kiwifruit genotypes A0134.41 and Ac174.46 (*A. chinensis* Planch. var. *chinensis*) were used for stable transformation studies. A0134.41 is a chance seedling introduced in 1993 from the Guangdong Academy of Agricultural Science, Guangzhou, China, and Ac174.46 is a male genotype selected from a controlled cross at the University of Udine.

Plant material was taken from mature individuals growing at the Experimental Farm “A. Servadei”, University of Udine. Shoot tips were sterilized by immersion in 70% ethanol for 20 sec and then in 1% sodium hypochlorite with three drops of Tween-20 for 20 min. Finally they were washed three times with sterile distilled water and cultured in vitro on MS substrate [38] supplemented with 0.2 mg/L IAA (indoleacetic acid), 2.5 mg/L BA (benzyl adenine), 2.5 % sucrose, 5 g/L glucose, and 7 g/L agar (B&V Parma).

### 4.2. Target Identification and CRISPR/Cas9 T-DNA Vector Assembly

A two-sites editing approach was adopted to increase gene knock-out efficiency. A panel of sgRNAs were designed to target different sites within the *SyGI* gene [17] using the software Cas-Designer [39]. Pairs of sgRNAs were evaluated in order to get a major deletion, in addition to single site-specific mutations. The sgRNA1 was selected among those located on the first exon of the gene, while sgRNA2 was identified among those mapping to the second exon (Figure 1). As reference genome for Cas-Designer software, we used the latest version of the *Actinidia chinensis* genome [40]. Cas-Designer also includes microhomology-predictor, which predicts the mutation patterns caused by microhomology-mediated end joining (MMEJ) pathway, and estimates how frequently unwanted in-frame deletions would happen. To perform the off-target analysis of the sgRNAs, we used the online tool Cas-OFFinder [41], integrated into Cas-Designer software, with default parameters. To select the in-silico best performing guides, we considered the following scores: i. GC contents ratio without PAM sequence equal to 50% for the first gRNA and 55% for the second one (20% to 80% is recommended); ii. out of frame score of 76 for the first gRNA and 72 for the second one (more than 66 is recommended); iii. a score equal to zero in the mismatch evaluation throughout the whole genome (meaning no off-target sites detected). The genome editing tool-kit system we used was a CRISPR/Cas9 multiplexing vector [24], able to clone multiple sgRNAs within an array controlled by an RNA Pol II promoter. That modular cloning system offers the possibility of producing a polycistronic transcript containing multiple gRNAs. The polycistronic RNA messenger is subsequently processed into single gRNAs by RNA-cleaving enzymes. We used the enzyme CRISPR-associated RNA endoribonuclease *Csy4* from *Pseudomonas aeruginosa* [42], provided by the cloning system. Among different plasmid solutions, we selected p*DIRECT22C* vector, which carries the T-DNA cassette for *Agrobacterium*-mediated transformation and is useful as a cloning backbone and template for PCR amplification of the two sgRNAs. The approach is based on Golden Gate assembly of PCR products, which carry the processing elements, gRNA repeats, as well as parts of target-specific gRNA spacers, providing unique sequence stretches used to design the Golden Gate junctions [24].

Three primer pairs were designed to amplify three desired regions of the vector cassette and integrate the sequence with restriction sites and the two sgRNAs. The primers used are reported in Appendix A. The first amplicon contains the *Sap*I restriction site, the *CmYLC* promoter, the *Csy4* repeat, and the first 12 nucleotides of the sgRNA1, and the *Esp*3I restriction site. The second amplicon contains the *Esp*3I site, the last 12 nucleotides of the sgRNA1, *Csy4* repeat, and the first 12 nucleotides of the sgRNA2, together with the *Esp*3I site. The last amplicon contains the *Esp*3I site, the last 12 nucleotides of the sgRNA2, the *Csy4* repeat, and the *Sap*I site. The obtained amplicons are shown in Appendix A. PCR reaction for each sgRNA cassette was set up using the proofreading DNA polymerase Q5 (NEB—New England Biolabs) as follows: 2.5 µL of each 10 µM stock primer (Forward and Reverse), 1 µL of 10 mM dNTPs, 0.02 U/µL of Q5 polymerase with the related 5X buffer, 1 ng of p*DIRECT22C* miniprep, to 50 µL final reaction volume. The PCR thermal conditions were 98 °C for 1 min, followed by 30 cycles at 98 °C for 10 sec and 60 °C for 15 sec and 72 °C for 15 sec, and a final extension at 72 °C for 2 min.

Owing to the presence of two *Csy4* repeats within the p*DIRECT22C* vector, two products can be amplified in the first reaction (when amplifying the promoter containing fragment), since the reverse primers binds to the repeated sequence. To prevent this, 500 ng of the p*DIRECT22C* vector was digested with 10 units of *Ban*I enzyme (NEB). That restriction enzyme cleaves the sgRNA repeat sequence and separates the two *Csy*4 repeats. The result of restriction was verified on 1% agarose gel, purified, and used as a template for the PCR reaction 1 (Appendix A). The following Golden Gate cloning step was set up in a 20-µL reaction as follows: 50 ng digested vector, 0.5 µL each PCR product diluted ten times, 5 U each restriction enzyme, *Sap*I and *Esp*3I (NEB), and 150 U/µL T7 DNA ligase (NEB) with the related 2X ligase buffer. The reaction runs for 10 cycles at 37 °C for 5 min and 25 °C for 10 min.

### 4.3. Final CRISPR/Cas9 T-DNA Vector Purification and Sequencing

The obtained p*DIRECT22C* carrying the two sgRNAs (p*DIRECT22C*:gRNAs) was cloned into *Escherichia coli* DH5α competent cells (NEB; [43]), using 5 µL of the Golden Gate reaction. After transformation, several colonies were screened through colony PCR with specific primers (reported in Appendix A) for the assembled vector flanking the region in which sgRNAs were inserted. One positive colony was grown overnight on LB liquid medium supplemented with 50 mg/L kanamycin, then miniprep was performed on the grown cultures to purify p*DIRECT22C*:gRNAs plasmid using Qiaprep^®^ Spin Miniprep Kit (Qiagen, Venlo, NL). DNA quantification was performed using both Nanodrop ND-100 spectrophotometer (ThermoFisher, Waltham, MA, USA) and Qubit^®^ 2.0 Fluorometer (Invitrogen, Carlsbad, CA). The purified plasmid was tested for the absence of putative contaminants by *Hind*III (NEB) restriction analysis and sequenced through NGS Illumina technology. “Celero™ DNA-Seq” kit (NuGEN, San Carlos, CA, USA) was used for library preparation following the manufacturer’s instructions. Both input and final libraries were quantified by Qubit^®^ 2.0 Fluorometer and quality tested by Agilent 2100 Bioanalyzer High Sensitivity DNA assay (Agilent Technologies, Santa Clara, CA, USA). The library was sequenced on the Illumina MiSeq instrument in paired-end 300-bp mode.

Pair-end raw reads were cleaned masking residuals of adapter sequences with Cutadapt tool [44], trimmed by quality, and filtered by possible contaminants using ENRE-Filter tool [45]. Resulting reads were assembled with the *de novo* assembler for short DNA sequence reads SSake [46] using default parameters. p*DIRECT22C*:gRNAs assembled sequence was aligned and compared to the predicted sequence using the pairwise sequence alignment tool EMBOSS Needle [47] with default parameters.

### 4.4. Actinidia Transformation

ElectoMAX^TM^
*Agrobacterium tumefaciens* LBA4404 cells (Invitrogen) and *A. tumefaciens* EHA105 chemically competent cells, prepared according to standard protocols, were transformed with the assembled p*DIRECT22C*:gRNAs vector. Colonies were screened through colony PCR with specific primers (Appendix A), and one positive colony was selected to be used to infect the male kiwifruit cultivars A0134.41 and Ac174.46, following the procedure described by Wang et al. and Wang and Lin-Wang [48,49]. The *Agrobacterium*-transformed strains re-suspended in half strength MS [38] basal medium and vitamins [50] liquid medium, supplemented with 2% sucrose and 100 µM acetosyringone at pH 5.8, were used to infect leaf discs and petioles from in vitro grown shoots, by co-cultivation for 30 min. The inoculated plant material was transplanted onto co-cultivation medium (Appendix A) and incubated at 25 ± 2°C for two days, then transferred to regeneration and selection medium (Appendix A) for four weeks at 25 ± 2°C and 16-h photoperiod, and refreshed every four weeks. Kanamycin-resistant adventitious buds initiated from the calli were excised individually and transferred to shoot elongation medium for four weeks, then transferred to shoot induction and maintenance medium (Appendix A).

### 4.5. Mutation Detection

Genomic DNA was extracted from leaf tissue of 20-week-old regenerated plants using a CTAB-based protocol [51]. DNA quantification was performed using the Nanodrop ND-100 spectrophotometer. p*DIRECT22C*:gRNAs vector integration within the regenerated plant genome was tested through PCR amplification, using the primers flanking the region in which sgRNAs were inserted in the vector (Appendix A).

The fragments encompassing each sgRNA target site were amplified using specific primers flanking the expected edit sites (detailed in Appendix A). PCR reactions were performed using the CloneAmp HiFi Polymerase (Takara, Shiga, Japan), following the manufacturer’s protocol, and the amplification success was verified running 5 µL of each PCR product on a 1.5% agarose gel. Amplicons were purified using the MinElute PCR purification kit (Qiagen). About 500 ng of each purified PCR product were analyzed by the T7EI assay, using T7 endonuclease I (NEB), according to the manufacturer’s instruction. The digestion products were analyzed by 1.5% agarose gel electrophoresis.

As for the sgRNA1, the related PCR products were also tested by *Bsl*I digestion, exploited as screening assay of the transformed plants thanks to sequence composition of the cleavage site targeted by sgRNA1. About 400 ng of each PCR product were digested with 10 U of *Bsl*I restriction enzyme (NEB) and checked by agarose gel electrophoresis. The same PCR products were sequenced through NGS Illumina technology using “Celero™ DNA-Seq” kit (NuGEN), as described previously (see “Final T-DNA vector purification and sequencing” paragraph). Validated Amplicon-seq libraries were sequenced on Illumina MiSeq platform yielding approximately 107,000 to 218,000 300-nt paired-end reads per sample, after quality check by FASTQC v0.11.4 [52].

Raw reads were adapter- and quality-trimmed using Trim Galore v0.4. (https://github.com/FelixKrueger/TrimGalore) and aligned to the *SyGI* gene reference sequence (LC482708.1 *Actinidia chinens*is DNA, Y-specific genomic marker fourth one from 45,725 bp to 46,151 bp in case of sgRNA1 and from 47,255 bp to 47,686 bp in case of sgRNA2 [18]) using Bowtie v2.2.6 [53]. Using custom scripts and the Samtools v1.3 suite [54], mapped reads were sorted into two major sequence variants based on eight single nucleotide polymorphisms at genomic position 45,901, 45,915, 45,922, 45,928, 45,936, 45,940, 45,953, and 45,971, respectively. Following analyses were restricted to reads matching the closest variant to the reference gene sequence. Overlapping paired-end reads originated from the same fragment were merged using Flash [55]. Detection of editing events was performed by analyzing merged reads with the CRISPResso software v2.0.32 with default settings [56].

## Figures and Tables

**Figure 1 plants-10-00062-f001:**
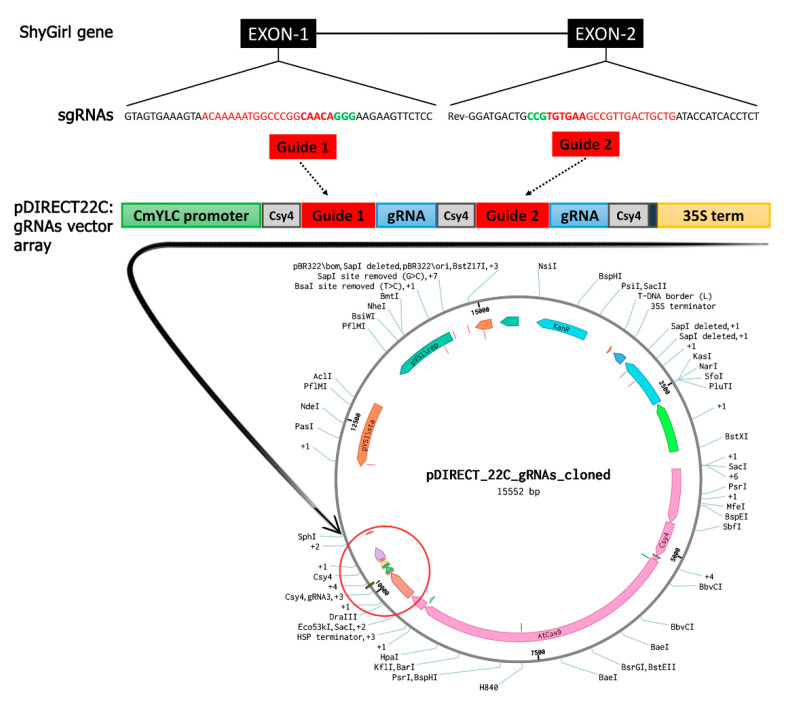
Diagram of the Cas9/sgRNA p*DIRECT22C* vector and target site selection in the *SyGI* gene. Above, the *SyGI* gene is schematically represented: the black boxes represent the two exons of the gene connected through the line, which represents the intron. The sequence of two sgRNAs selected on the first and second exon are reported in red. In the center, the structure of the final p*DIRECT22C*:gRNA vector cassette is represented with the two cloned sgRNAs (red boxes): the green box represents the *CmYLC* promoter, the three grey boxes represent the *Csy4* repeat, blue boxes represent the gRNA repeat sequences, and the yellow box represents the 35S terminator. Below, the final p*DIRECT22C*:gRNA vector containing 35S:*Csy4*-P2A-AtCas9 + *CmYLCV*:gRNAs with *Csy4* spacers, Plant Selection: 2x35S:npt II. The sgRNA cloning site is circled in red.

**Figure 2 plants-10-00062-f002:**
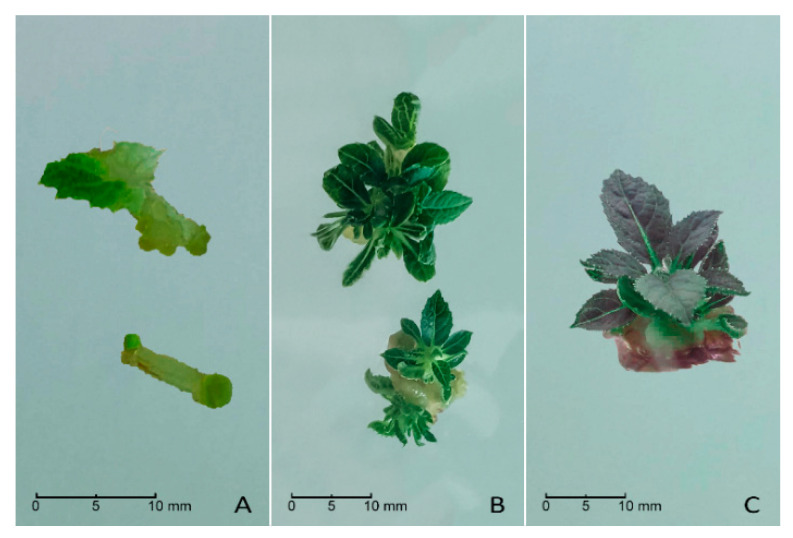
In vitro cultures (Ac174.46 genotype) after co-cultivation with Agrobacterium strains containing p*DIRECT22C*:gRNAs. (**A**) Kanamycin-resistant calli appeared along the cut petioles in 4–8 weeks. (**B**) Kanamycin-resistant adventitious buds regenerated from calli in 12–16 weeks. (**C**) Putative transgenic shoots ready for in vitro propagation on elongation medium before genetic analyses.

**Figure 3 plants-10-00062-f003:**
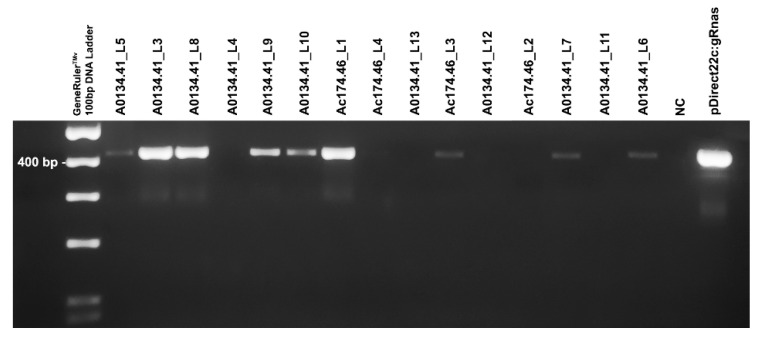
PCR detection of p*DIRECT22C*:gRNAs vector integration within the genome of the 11 putative transgenic shoots obtained for the male genotype A0134.41 and four putative transgenic shoots obtained for the Ac174.46 cultivar. The primer pair TC320/TC089R amplifies the p*DIRECT22C*:gRNA vector cassette where the two guideRNAs were cloned, producing a 418 bp fragment. GeneRuler^TM^ 100 bp DNA Ladder (Thermo Fisher) was used for sizing the PCR products in the range of 100 to 1000 bp on agarose gel. Plant line code of each putative transgenic shoot is reported. The last two samples represent the negative control (NC, Milli-Q water) and the positive control (the p*DIRECT22C*:gRNAs vector), respectively. Image for illustrative purpose. The original image is reported in Appendix A.

**Figure 4 plants-10-00062-f004:**
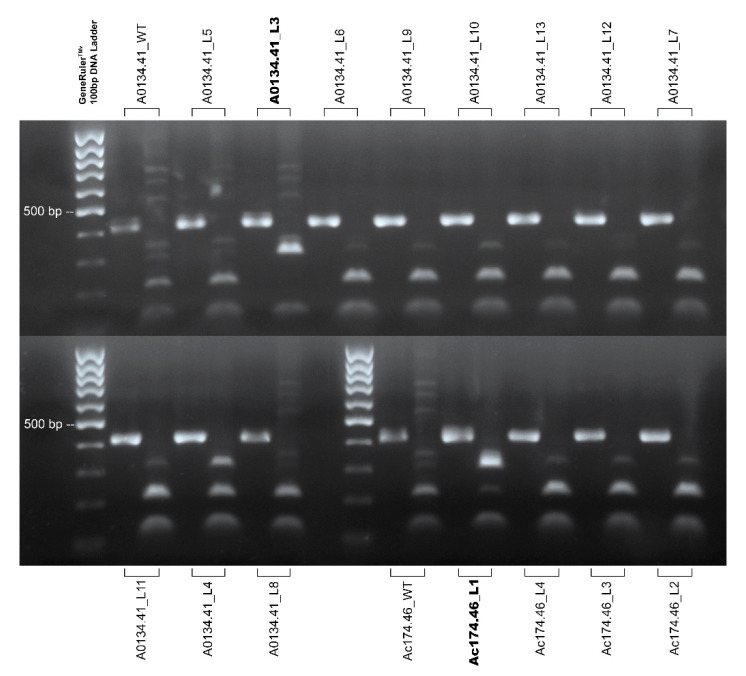
Agarose gel electrophoresis of the sgRNA1 target sequence PCR indigested product and their *Bsl*I digestion. The male genotype A0134.41 *wild-type* (WT), plus the 11 putative transgenic shoots obtained from this male genotype, are displayed in the upper panel of the figure and in the first three rows of the bottom panel of the figure. The male genotype Ac174.46 *wild-type* (WT) plus the four putative transgenic shoots obtained from this male genotype are displayed in the bottom panel of the figure. GeneRulerTM 100 bp DNA Ladder (Thermo Fisher) was used for sizing the PCR products in the range of 100 to 1000 bp on agarose gel. The plant line code of each sample is reported. The amplicon of the sgRNA1 target assay is 427 bp in length, and its *Bsl*I digestion produces three fragments of 221, 98, and 108 bp (the latter two bands appear as one). PCR amplification of some lines, i.e., A0134.41, *wild-type* (WT), A0134.41_L5, and Ac174.46 *wild-type* (WT), co-amplifies a second sequence that results in the presence of two extra-bands in the restriction pattern. Thirteen of these plants exhibited a restriction pattern compatible with the *wild-type*, showing a fragment of 221 bp. The same assay on an edited sequence provides a different digestion pattern, producing two fragments (320 and 108 bp). The edited lines are reported in bold. Image for illustrative purpose. The original image is reported in Appendix A.

**Figure 5 plants-10-00062-f005:**
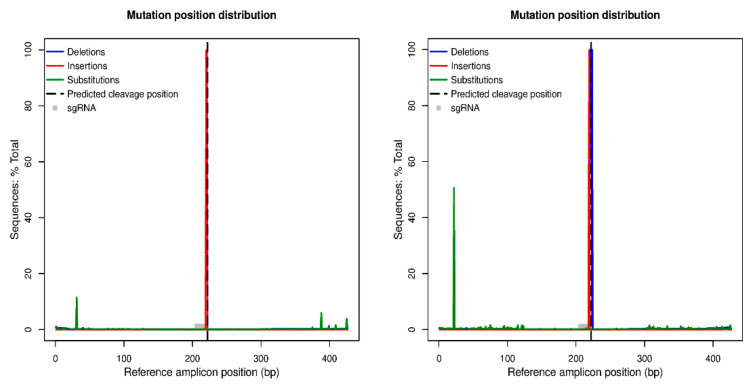
CRISPResso output analysis for the two regenerated lines named A0134.41_L3 (on the left) and Ac174.46_L1 (on the right). The two plots allow the quantification and visualization of the position and type of outcomes within the amplicon sequence reporting the frequency of mutations detected across sgRNA1 *locus* in this case. The reference amplicon position in the base pair is reported on the X-axis, while the Y-axis reports the percentage of reads showing mutations, with the effective number of mutations in brackets. Mutation types are color-coded: insertions are in red, deletions are purple, and substitutions are green.

**Figure 6 plants-10-00062-f006:**
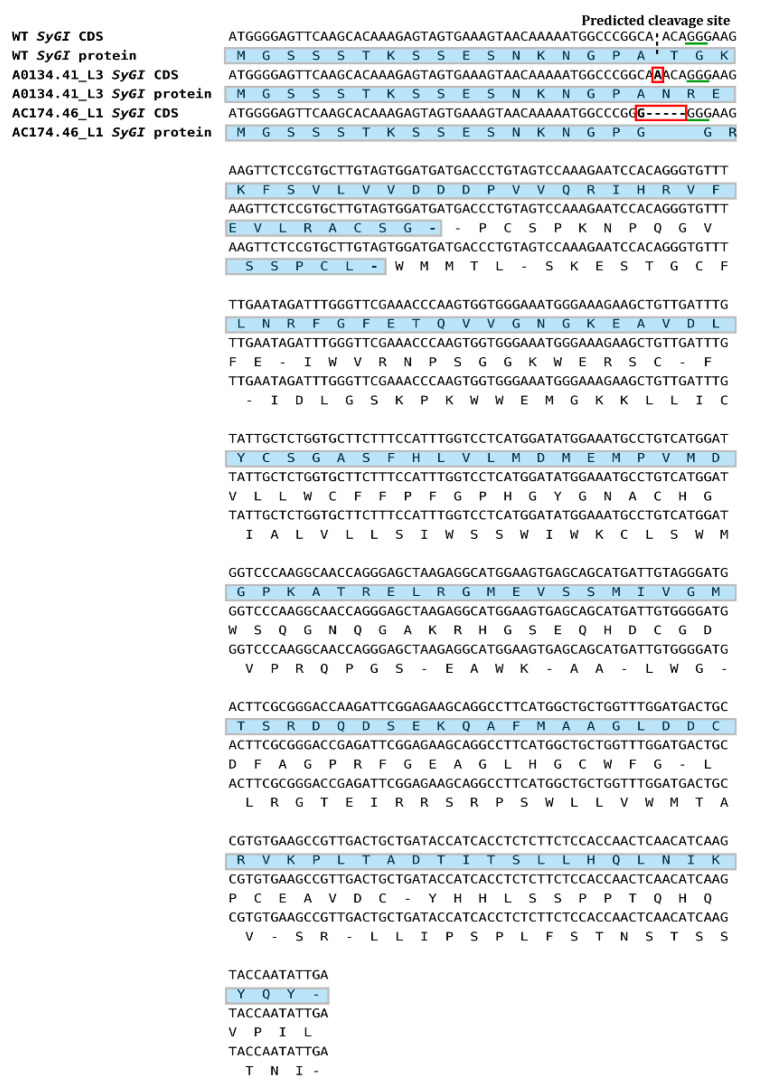
Alignment of coding sequences (CDS) of *Wild-Type SyGI* (WT) and edited *SyGI* in the A0134.41_L3 and Ac174.46_L1 lines. PAM sequence is underlined in green. In the WT *SyGI* CDS, the predicted cleavage site is underlined by a stuttered line, and it has been found three bases upstream of the PAM sequence. The insertion of a single A in plant line A0134.41_L3 and the deletion of five bases plus the insertion of a single G in plant line Ac174.46_L1 are underlined in red. Protein translation is shown; in both edited lines, the modifications cause a reading frame shift, with a consequent formation of a premature stop codon.

**Figure 7 plants-10-00062-f007:**
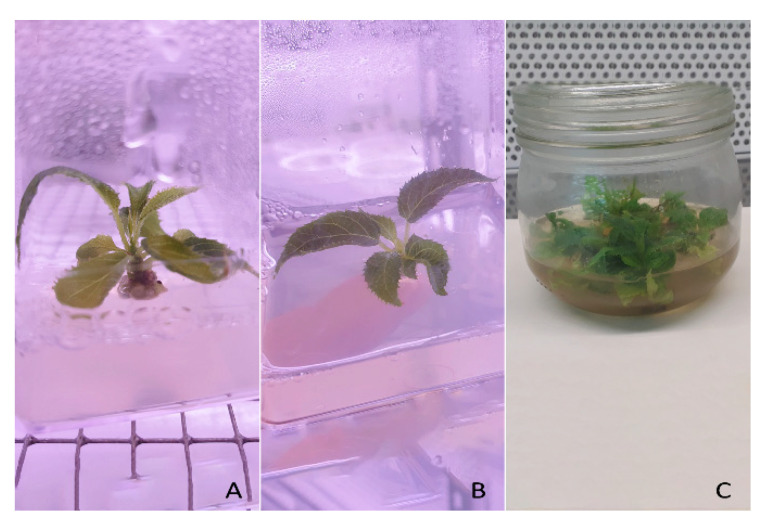
In vitro culture of the regenerated plantlets. (**A**) Non-edited line A0134.41_L8 on elongation medium. (**B**) Edited line Ac174.46_L1 on elongation medium. (**C**) Edited line A0134.41_L3 propagated on shoot induction and maintenance medium.

**Table 1 plants-10-00062-t001:** Summary of *Actinidia* transformation.

Genotype	No. of Leaf Discs	No. of Petioles	Kanamycin-Resistant Calli		No. of Putative Transgenic Shoots	No. of Tested Plantlets
			No. of Leaf Discs	No. of Petioles		
A0134.41	40	35	2	30	44	11
Ac174.46	50	33	3	20	61	4

**Table 2 plants-10-00062-t002:** Summary of the genetic evaluation for the putative transgenic regenerated line.

Genotype	Line Code	Ti integration *	*Bsl*I Restriction Assay **	Illumina Sequencing
A0134.41	A0134.41_L5	Yes	Site-specific digestion and *wild-type* patterns	0.0038% edited reads
A0134.41	A0134.41_L6	Yes	Site-specific digestion and *wild-type* patterns	0.20% edited reads
A0134.41	A0134.41_L3	Yes	Site specific digestion	99.85% edited reads
A0134.41	A0134.41_L8	Yes	*Wild-type* digestion pattern	0.00% edited reads
A0134.41	A0134.41_L4	No	Site-specific digestion and *wild-type* patterns	0.091% edited reads
A0134.41	A0134.41_L9	Yes	*Wild-type* digestion pattern	0.00% edited reads
A0134.41	A0134.41_L10	Yes	Site-specific digestion and *wild-type* patterns	0.31 % edited reads
Ac174.46	Ac174.46_L1	Yes	Site specific digestion	99.76% edited reads
Ac174.46	Ac174.46_L4	No	*Wild-type* digestion pattern	Not sequenced
A0134.41	A0134.41_L13	No	*Wild-type* digestion pattern	Not sequenced
Ac174.46	Ac174.46_L3	Yes	*Wild-type* digestion pattern	Not sequenced
A0134.41	A0134.41_L12	No	*Wild-type* digestion pattern	Not sequenced
Ac174.46	Ac174.46_L2	No	*Wild-type* digestion pattern	Not sequenced
A0134.41	A0134.41_L7	Yes	*Wild-type* digestion pattern	Not sequenced
A0134.41	A0134.41_L11	No	*Wild-type* digestion pattern	Not sequenced

* PCR amplification results of *Ti* integration for p*DIRECT22C:*gRNAs vector are shown in Figure 3. ** PCR amplification and restriction results of the assay are shown in Figure 4.

## Data Availability

The data presented in this study are available within the article and supplementary materials. The PCR products raw reads data are available on request from the corresponding author.

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
