# Peer review of "Targeted Mutagenesis of the Female-Suppressor SyGI Gene in Tetraploid Kiwifruit by CRISPR/CAS9"

_plants, 2020, doi:10.3390/plants10010062_

Round 1
Reviewer 1 Report
I am statistician and for me this paper is good. This is not my field of work.
Author Response
We wish to thank the reviewer number one for appreciating our work. We have not made any changes considering that referee number one did not express any comments about our manuscript.
Reviewer 2 Report
The manuscript entitled “Targeted mutagenesis of the female-suppressor SyGI gene in tetraploid kiwifruite by CRISPR/CAS9” by Dr. De Mori et al., describes the editing of the SyGI gene on the tetraploid male kiwifruit genotypes.
Recent molecular works on sex-determining genes on the Y chromosome in kiwifruit demonstrated that two genes named Shy Girl (SyGI) and Friendly boy (FrBy) act as the suppressor of female development and the male promoting factor (M factor), respectively. In the case of FrBy, expression of FrBy in female kiwifruit resulted in hermaphrodite plants, indicating that FrBy acts as the M factor in kiwifruit sex determination. On the other hand, Akagi et al. has shown recently that the loss of SyGI resulted in a natural hermaphrodite line. However, knock-out experiments of SyGI have not been previously reported. In this paper, the authors exploited CRISPR/Cas9 system to knock-out the SyGI gene in kiwifruit and obtained two regenerated knock-out lines.
The manuscript describes the successful knock-out of SyGI gene in detail, and the results are promising. However, the authors did not provide the phenotypic evaluation of the flowers due to the long time needed to develop flowering plants. Thus, the manuscript lacks scientific novelty. I want to ask the authors to include the phenotypic evaluation of the knock-out plants. Akagi et al. used “rapid flowering” kiwifruit lines that flowered four months after regeneration (Akagi et al., Nature Plants 5, 801, 2019). Please use the similar “rapid flowering” lines as discussed in lines 389-394 in the text, or include the phenotypic evaluation of non-flower organs of the knock-out plants described in the manuscript (A0134.41_L3 and Ac174.46_L1 lines).
It would be helpful for readers if the authors provide photos of two edited lines. I wonder whether the edited lines grow normally and look healthy.
Figure 1. The characters are too small. Please magnify the figure a little bit.
Author Response
REVIEWER 2
Comments and Suggestions for Authors:
The manuscript entitled “Targeted mutagenesis of the female-suppressor SyGI gene in tetraploid kiwifruite by CRISPR/CAS9” by Dr. De Mori et al., describes the editing of the SyGI gene on the tetraploid male kiwifruit genotypes.
Recent molecular works on sex-determining genes on the Y chromosome in kiwifruit demonstrated that two genes named Shy Girl (SyGI) and Friendly boy (FrBy) act as the suppressor of female development and the male promoting factor (M factor), respectively. In the case of FrBy, expression of FrBy in female kiwifruit resulted in hermaphrodite plants, indicating that FrBy acts as the M factor in kiwifruit sex determination. On the other hand, Akagi et al. has shown recently that the loss of SyGI resulted in a natural hermaphrodite line. However, knock-out experiments of SyGI have not been previously reported. In this paper, the authors exploited CRISPR/Cas9 system to knock-out the SyGI gene in kiwifruit and obtained two regenerated knock-out lines.
The manuscript describes the successful knock-out of SyGI gene in detail, and the results are promising. However, the authors did not provide the phenotypic evaluation of the flowers due to the long time needed to develop flowering plants. Thus, the manuscript lacks scientific novelty. I want to ask the authors to include the phenotypic evaluation of the knock-out plants. Akagi et al. used “rapid flowering” kiwifruit lines that flowered four months after regeneration (Akagi et al., Nature Plants 5, 801, 2019). Please use the similar “rapid flowering” lines as discussed in lines 389-394 in the text, or include the phenotypic evaluation of non-flower organs of the knock-out plants described in the manuscript (A0134.41_L3 and Ac174.46_L1 lines).
It would be helpful for readers if the authors provide photos of two edited lines. I wonder whether the edited lines grow normally and look healthy.
Figure 1. The characters are too small. Please magnify the figure a little bit.
We wish to thank the reviewer number two for providing valuable and constructive comments. Please find here below our response:
We did not use the “rapid flowering” kiwifruit line described by Varkonyi‐Gasic et al. 2019 in our experiment because this line is a female kiwifruit genotype and our target was the knock-out of SyGI gene which is only present in male kiwifruit plants, as described by Akagi et al. 2018. A rapid flowering male genotype, as far as we know, is not available yet.
Moreover, in contrast to the previous publications, where diploid genotypes were investigated, we conducted our experiment using two tetraploid male genotypes. We were interested in these genotypes because of their frequent use as pollen donors in our breeding programs. As stated in the manuscript, in dioecious plants, the pollen is usually selected in the absence of knowledge on the genetic background controlling the fruit trait, as fruiting characteristics are not expressed in males. We hope to evaluate all these aspects in the future, once phenotypic effects caused by the knock-out of SyGI gene will be visible.
We accepted the suggestion of the referee about the figures. We modified the figure number 1 by increasing the font size. We added a new figure (Figure 7) showing no edited and edited genotypes. Plantlets are growing normally and look healthy.
Reviewer 3 Report
The reviewer agrees with the changes. The title of the manuscript in this edition reflects its compliance with the experimental data presented.
Author Response
We wish to thank the reviewer number three for appreciating our work.
Reviewer 4 Report
Review of the paper by G.D. Mori and colleagues “Targeted mutagenesis of the female-suppressor SyGI gene in tetraploid kiwifruit by CRISPR/Cas9”
for Plants; Manuscript ID: plants-993212.
Reviewer's report
The present paper reports about the application of the powerful CRISPR/Cas9-based gene editing technology to knock-out the female suppressor gene SyGI in kiwifruit. The CRISPR/Cas9 gene editing is very useful in forest and fruit trees, because this technology can accelerate tree breeding in an undreamed manner.
Sex-determination in kiwifruit has been described impressively by Akagi et al. (2018, 2019) as a two-gene-system, by having SyGI (a C-type cytokinin response regulator) acting as suppressor of female development (by repressing SuF, the feminisation gene), and FrBy acting as male promoting factor. Overexpression of SyGI suppresses female development, and knockout of FrBy leads to hermaphrodite flowers. Knockout (not published so far) of SyGI in male kiwifruit is expected to also lead to hermaphrodite flowers.
Unfortunately, the present manuscript doesn’t provide any answers on the very interesting questions raised, because of the long-vegetative period of kiwifruit and the long time span this species needs to flower for the first time. Although, as described below, either appropriate early-flowering plant lines are available, or at least the protocol to knockout the Centroradialis gene to produce early-flowering genotypes has already been published.
A similar approach has been published for poplar quite recently in Nature plants, June issue, 2020. Here, the A-type cytokinin response regulator ARR17 has been knocked-out by CRISPR/Cas9 in female early-flowering poplar. As a result, the ARR17-knocked-out female early-flowering poplar produced male flowers!
I cannot recommend to publishing this manuscript in its current version. The reasons for my decision are given below:
- As far as I know is it unusual to include a reference in the Abstract
- The first sentence of the abstract claims that all 54 species of Actinidia are dioecious. But the authors their selves restrict in the Introduction that dioecy has really been demonstrated just for few Actinidia species. Therefore, the first sentence in the Abstract has to be modified accordingly.
- The same can be assigned for the statement in the second sentence of the Abstract, namely that all Actinidia species operate with XX/XY-system. It cannot be excluded that one of the “dioecy-unknown” Actinidia species operate with ZW/ZZ.
In the genus Populus, all species with known sex determination species have XX/XY with only one exception, as far as it is known, Populus alba, having ZW/ZZ. - The authors state in the Abstract that they are not able to verify phenotypic effects caused by the knockout of SyGI due to long vegetative phases of kiwifruit. The same it is mentioned in the Introduction and in the Discussion. In the Discussion, the authors mention a transgenic kiwifruit Centroradialis mutant characterized by early flowering. Also, Akagi et al. have used a “rapid flowering” A. chinensis cultivar in their experiments, what is very clever. I’m wondering why the authors didn’t use this cultivar for their CRISPR-knockout of SyGI to unravel its function in sex determination?
- I also miss in the Introduction the state-of-the-art of kiwifruit transformation and CRISPR/Cas9 description (which is described in the Discussion). Instead the authors mention very general “mutant creation” and “targeted mutagenesis”.
- Due to the circumstance that the authors are not able to present any phenotypic data which confirm the effect of SyGI-knockout, they report about SyGI target identification, vector assembly, Actinidia transformation, and CRISPR-mutant detection in a very extensive manner.
- SyGI target identification: no references are given for Cas-Designer (line 97) and Cas-OFFinder (line 110), just at the end in Material and Methods, the references are mentioned. This part can be reduced.
- Vector assembly: This part has to be moved in the Material and Methods section
- Actinidia transformation: the authors describe kiwifruit transformation procedure very extensive in the way as if it is the very first kiwifruit transformation description. However, the first time that kiwifruit has been transformed is more than 25 years ago. Ok, the two genotypes used in this study are transformed for the first time, thus, this can be described in maximal five sentences.
Please add (A) and (B, C) following “Figure 2” at the respective positions. - CRISPR-mutant detection: also this part can be reduced significantly. Ok, a fewer number of publications on successful CRISPR/Cas9 application in kiwifruit has been published than transformation protocols, but there are some publications available. And the way how the authors have analysed the putative mutants is very circuitous. I fully can understand that a PCR-RFLP analysis is helpful when hundreds of samples have to be analysed, but the authors have obtained 105 putative transgenic shoots in total, and out of these, they selected just 15! So instead of applying PCR-RLFP, I would favor to Sanger-sequence the 15 samples by using the primer-pair used for PCR. Sanger sequences do then also make Illumina-sequencing superfluous.
- The quality of the two figures 3 and 4, as shown in the Supplementary material, is very bad. Many weak extra bands, just one time a size information of the bands in the molecular ladder, no information about name of the ladder and company. In the legend, it is mentioned a “negative control” without mentioning the term “CN” (why not “NC”?), and what I don’t like is to position the plant codes within the gel (unspecific bands are hided) but these should be placed on the top.
And why didn’t include the authors a non-transgenic control, especially in Figure 4 this would be a real control for restriction pattern (as mentioned in line 209). In Table 2, the authors claim “wild-type patterns” but without showing these.
And finally, I miss (A) and (B) for the upper and lower part of the Figure 4. - In Table 2, I don’t understand the results for A0134.41_I.9, which show “Site-specific digestion plus wild-type patterns” (as e.g. it has been also mentioned for A0134.41_I.4 or A0134.41_I.10, however with 0.00% edited reads? How do the authors explain this result?
- In lines 386ff, the authors mention that they also have to analyse more putative transgenic lines. And in lines 388ff, they claim, that it will take some time to assess the phenotypic effects of SyGI knockout. Yes, I fully agree with both, let’s wait for these results.
Author Response
REVIEWER 4
We wish to thank the reviewer number four for providing valuable and constructive comments. Please find here below our response:
As far as I know is it unusual to include a reference in the Abstract
We accepted the suggestion of the reviewer and removed the reference from the abstract.
The first sentence of the abstract claims that all 54 species of Actinidia are dioecious. But the authors their selves restrict in the Introduction that dioecy has really been demonstrated just for few Actinidia species. Therefore, the first sentence in the Abstract has to be modified accordingly.
We accepted the suggestion of the referee and modified the abstract accordingly.
The same can be assigned for the statement in the second sentence of the Abstract, namely that all Actinidia species operate with XX/XY-system. It cannot be excluded that one of the “dioecy-unknown” Actinidia species operate with ZW/ZZ.
In the genus Populus, all species with known sex determination species have XX/XY with only one exception, as far as it is known, Populus alba, having ZW/ZZ.
We accepted the suggestion of the referee and removed the assertion about sex determination behavior in all Actinidia taxa.
The authors state in the Abstract that they are not able to verify phenotypic effects caused by the knockout of SyGI due to long vegetative phases of kiwifruit. The same it is mentioned in the Introduction and in the Discussion. In the Discussion, the authors mention a transgenic kiwifruit Centroradialis mutant characterized by early flowering. Also, Akagi et al. have used a “rapid flowering” A. chinensis cultivar in their experiments, what is very clever. I’m wondering why the authors didn’t use this cultivar for their CRISPR-knockout of SyGI to unravel its function in sex determination?
The issue raised by the reviewer is accurate. However, Akagi et al. 2019 used the “rapid flowering” A. chinensis cultivar obtained from the previous editing work of Varkonyi‐Gasic et al. 2019. We did not use the “rapid flowering” kiwifruit line in our experiment because that line is a female kiwifruit genotype and our target was the knock-out of SyGI gene which is only present in male kiwifruit plants, as described by Akagi et al. 2018. A rapid flowering male genotype, as far as we know, is not available yet.
Moreover, in contrast to the previous publications, , where diploid genotypes were investigated we conducted our experiment using two tetraploid male genotypes. We were interested in these two genotypes because of their frequent use as pollen donors in our breeding programs. As stated in the manuscript, in dioecious plants, the pollen is usually selected in the absence of knowledge on genetic background controlling the fruit trait, as fruiting characteristics are not expressed in males. All these aspects will be evaluated in the future, once phenotypic effects caused by the knock-out of SyGI gene will be visible.
I also miss in the Introduction the state-of-the-art of kiwifruit transformation and CRISPR/Cas9 description (which is described in the Discussion). Instead the authors mention very general “mutant creation” and “targeted mutagenesis”.
We accepted the suggestion, and updated the introduction including the state-of-the-art of the kiwifruit editing using CRISPR/Cas9 technique.
SyGI target identification: no references are given for Cas-Designer (line 97) and Cas-OFFinder (line 110), just at the end in Material and Methods, the references are mentioned. This part can be reduced.
We removed from paragraph 2.1 in Results section some details already described in the Material and Methods section.
Vector assembly: This part has to be moved in the Material and Methods section
We removed the first part of the paragraph, accepting the referee suggestion.
Actinidia transformation: the authors describe kiwifruit transformation procedure very extensive in the way as if it is the very first kiwifruit transformation description. However, the first time that kiwifruit has been transformed is more than 25 years ago. Ok, the two genotypes used in this study are transformed for the first time, thus, this can be described in maximal five sentences
.Please add (A) and (B, C) following “Figure 2” at the respective positions.
We fully agree, thus we shortened the description of the plant transformation procedure (see paragraph 4.4). We also added (A), (B) and (C) at the respective positions as suggested by the reviewer.
CRISPR-mutant detection: also this part can be reduced significantly. Ok, a fewer number of publications on successful CRISPR/Cas9 application in kiwifruit has been published than transformation protocols, but there are some publications available. And the way how the authors have analysed the putative mutants is very circuitous. I fully can understand that a PCR-RFLP analysis is helpful when hundreds of samples have to be analysed, but the authors have obtained 105 putative transgenic shoots in total, and out of these, they selected just 15! So instead of applying PCR-RLFP, I would favor to Sanger-sequence the 15 samples by using the primer-pair used for PCR. Sanger sequences do then also make Illumina-sequencing superfluous.
The quality of the two figures 3 and 4, as shown in the Supplementary material, is very bad. Many weak extra bands, just one time a size information of the bands in the molecular ladder, no information about name of the ladder and company. In the legend, it is mentioned a “negative control” without mentioning the term “CN” (why not “NC”?), and what I don’t like is to position the plant codes within the gel (unspecific bands are hided) but these should be placed on the top.
And why didn’t include the authors a non-transgenic control, especially in Figure 4 this would be a real control for restriction pattern (as mentioned in line 209).
In Table 2, the authors claim “wild-type patterns” but without showing these.
And finally, I miss (A) and (B) for the upper and lower part of the Figure 4.
In Table 2, I don’t understand the results for A0134.41_I.9, which show “Site-specific digestion plus wild-type patterns” (as e.g. it has been also mentioned for A0134.41_I.4 or A0134.41_I.10, however with 0.00% edited reads? How do the authors explain this result?
We accepted the suggestion of the reviewer about the figures and we modified the figure number 3 adding the name of the ladder and company, we replaced the term “CN” with the more appropriate term “NC”, and we mentioned this term in the legend. Moreover, we moved the position of the plant codes on the top.
We apologize for the omission of more complete information about the complicated pattern of the PCR-RFLP analysis. We propose a new Figure 4 which better explains our results. The primer pair, used to amplify the 427-bp genomic target of sgRNA1 target region, has shown to co-amplify in a minor percentage a second sequence that diverges from the target by eight SNPs and a 10-nt deletion, as revealed by Illumina sequencing of target amplicons. As described in Discussion, we noticed that such sequence mapped to the Actinidia chinensis genome in the minus strand of LG19 at position 5,058,376, which is the locus of Achn384741 gene, i.e. the autosomic counterpart of SyGI gene. Consequently, PCR products are a mix of sequences belonging to the two loci, which, once digested using the BslI restriction enzyme, show a more complex digestion pattern than expected. Therefore, we prepared a new Figure 4 with the addition of the two non-transgenic controls, as requested by the reviewer, and an increased gel resolution in order to better show those aspects. Undoubtedly, the aim of the restriction assay is to discriminate easily and quickly edited plantlets from non-edited ones. To this regard, BslI restriction allows to identify putative edited plantlets thanks to the absence of the 221-bp fragment, which disappears when editing takes place.
This scenario allows to explain the results for A0134.41_L9 which has 0,00% edited reads. The wording “Site-specific digestion plus wild-type patterns” is actually incorrect and we edited text in Table 2 accordingly.
This scenario also allows to explain why the Sanger sequencing approach suggested by the reviewer is not suitable in this case. The mix of different sequences originating from the different loci would make the electropherograms analysis too difficult to deconvolute due to the overlapping of different signals. Moreover, Sanger sequencing would not have allowed to quantify, sort and disregard the sequences belonging to the second locus. In contrast, Illumina sequencing allowed us to observe the polymorphisms in the sequences and exclude the possibility of Achn384741 gene acting as an off-target in the editing procedure because of the presence of SNPs in the seed part of the sgRNA sequence, making Cas9 binding unlikely.
Therefore, in our case we believe that BslI restriction assay is a first good test to discriminate edited plantlets. We would like to thank the reviewer for showing us the work conducted on the ARR17 gene in poplar that, being an A-type cytokinin response regulator, shows functional similarity with the SyGI gene. As regard, we cited the work of Müller et al. 2020 in line 396-400. Further analysis will be conducted to investigate if a similar scenario could be present in the kiwifruit genome.
Round 2
Reviewer 4 Report
Second review of the paper by G.D. Mori and colleagues “Targeted mutagenesis of the female-suppressor SyGI gene in tetraploid kiwifruit by CRISPR/Cas9”
for Plants; Manuscript ID: plants-993212.
Reviewer's report
I review this paper now for the second time. The authors have considered all my comments. Although I feel that presence of a flowers in the edited lines would significantly increase the value of this paper, I accept the reasons explained by the authors why this is not so easy to implement. In addition, I don’t detect any pitfalls in this paper, so the manuscript can be published in its current version.
I don’t agree with the argumentation why Sanger sequencing is too complicate. Its nothing than the choose of appropriate primer pairs differentiating the alleles. But I agree that PCR-RFLP is a quick and low-cost alternative.
This manuscript is a resubmission of an earlier submission. The following is a list of the peer review reports and author responses from that submission.
Round 1
Reviewer 1 Report
The manuscript under review is devoted to the problem of overcoming dioeciousness in such an important agricultural crop as kiwi. The authors used in their experimental works the genomic editing method to introduce knockout mutations in the region of the SyGI gene, described as a suppressor of feminization. The authors used tetraploid kiwi lines for genomic editing and developed a CRISPR / Cas9 strategy for them using the pDIRECT22C vector. The paper presents examples of two events of successful editing of the selected target region of the target gene. The provided evidence is beyond doubt. In the presented form, the article can be recommended for publication in Plants journal, subject to its title change. The title of this article is aimed at providing experimental examples of genomic editing for the creation of hermaphrodite plants in kiwis (“Overcome dioecism in tetraploid kiwifruit: a CRISPR/CAS9 editing approach”). However, it is not. This article deals only with the induction of knockouts for the SyGI gene and the description of two regenerant lines without describing the phenotypic effects. The authors refer to the fact that they need a lot of time for this work, since kiwi plants do not develop as quickly as, for example, A.thaliana. Based on this, the reviewer recommends to reduce the title of this manuscript to the problem of obtaining knockouts for the SyGI gene in tetraploid kiwi lines and, in accordance with this, to make some changes in the text of the manuscript. Experimental data on the analysis of dioecious changes in the two resulting knockout lines, as well as in a larger number of them (which is also announced for the future in this manuscript), will serve as the subject for another publication related to overcoming kiwi dioeciousness.